# Predictive Utility of Biochemical Markers for the Diagnosis and Prognosis of Gestational Diabetes Mellitus

**DOI:** 10.3390/ijms252111666

**Published:** 2024-10-30

**Authors:** Sathaphone Inthavong, Phudit Jatavan, Theera Tongsong

**Affiliations:** Department of Obstetrics and Gynecology, Faculty of Medicine, Chiang Mai University, Chiang Mai 50200, Thailand; sathaphone.itv@gmail.com (S.I.); theera.t@cmu.ac.th (T.T.)

**Keywords:** adipokines, biomarker, gestational diabetes mellitus, inflammatory marker, insulin resistance, lipid profile

## Abstract

Gestational diabetes mellitus (GDM) is a common complication during pregnancy with an increasing prevalence worldwide. Early prediction of GDM and its associated adverse outcomes is crucial for timely intervention and improved maternal and fetal health. The objective of this review is to provide a comprehensive summary of contemporary evidence on biomarkers, focusing on their potential to predict the development of GDM and serve as predictors of maternal, fetal, and neonatal outcomes in women with GDM. A literature search was conducted in the PubMed database using relevant terms. Original research articles published in English between 1 January 2015, and 30 June 2024, were included. A two-stage screening process was employed to identify studies on biomarkers for GDM diagnosis and prognosis and to evaluate the evidence for each biomarker’s diagnostic performance and its potential prognostic correlation with GDM. Various biochemical markers, including adipokines, inflammatory markers, insulin resistance markers, glycemic markers, lipid profile markers, placenta-derived markers, and other related markers, have shown promise in identifying women at risk of developing GDM and predicting adverse pregnancy outcomes. Several promising markers with high predictive performance were identified. However, no single biomarker has demonstrated sufficient accuracy to replace the current diagnostic criteria for GDM. The complexity of multiple pathways in GDM pathogenesis highlights the need for a multi-marker approach to improve risk stratification and guide personalized management strategies. While significant progress has been made in GDM biomarker research, further studies are required to refine and validate these markers for clinical use and to develop a comprehensive, evidence-based approach to GDM prediction and management that can improve maternal and child health outcomes.

## 1. Introduction

Gestational diabetes mellitus (GDM) is a common complication during pregnancy, characterized by glucose intolerance that is first identified during the second or third trimester [1]. The prevalence of GDM ranges from 9.3% to 25.5% of pregnancies, depending on the ethnicity of the studied population and the diagnostic criteria used [2,3,4]. Globally, the occurrence of diabetes during pregnancy is rising due to increasing rates of obesity, type 2 diabetes mellitus (T2DM), and advancing maternal age [5,6,7]. GDM is associated with various adverse pregnancy outcomes, including macrosomia, preeclampsia, cesarean delivery, preterm birth, shoulder dystocia, birth trauma, neonatal hypoglycemia, neonatal hyperbilirubinemia, neonatal respiratory distress syndrome, and an increased likelihood of neonatal intensive care unit admission [8,9,10,11,12]. Exposure to hyperglycemia during pregnancy may also lead to long-term adverse effects on both women and newborns. Women with GDM have an increased risk of developing type 2 diabetes (T2DM), cardiovascular disease, hypertension, metabolic disorders, and renal disease [13,14,15,16,17,18]. Similarly, offspring of women with GDM are at increased risk of developing type 2 diabetes (T2DM), hypertension, obesity, metabolic syndrome, non-alcoholic fatty liver disease, and potentially autism later in life [19,20,21,22,23,24,25,26,27]. Understanding the pathophysiology of GDM is crucial for developing effective prevention and treatment strategies to mitigate the short- and long-term adverse effects on maternal and fetal health.

The progression of GDM involves a combination of insulin resistance and pancreatic β-cell dysfunction caused by various factors, including placental hormones, inflammation, metabolic changes, genetics, the environment, and epigenetic modifications [28]. Placental hormones, including progesterone, estrogen, cortisol, and human placental growth hormone, induce insulin resistance to ensure adequate glucose supply to the fetus [29,30,31,32]. However, insulin resistance becomes more severe in women with pre-existing conditions like obesity or polycystic ovary syndrome (PCOS) [33]. Simultaneously, pancreatic beta-cells may fail to adequately increase insulin production to meet the increased demand due to genetic factors, metabolic stress, and lipotoxicity [28,34,35]. Chronic low-grade inflammation, arising from adipose tissue and the placenta, further exacerbates insulin resistance and impairs beta-cell function through the action of pro-inflammatory cytokines [36,37]. Pregnancy-induced metabolic adaptations, including increased circulating lipids and altered adipokine levels, can also contribute to insulin resistance [38,39]. Environmental and lifestyle factors, such as diet and physical activity, play a significant role in modulating GDM risk [40,41]. Recent studies suggest that epigenetic changes affecting genes involved in glucose metabolism and insulin signaling may play a role in the pathogenesis of GDM [42,43]. The complexity of GDM pathogenesis underscores the importance of thorough screening and management approaches.

Gestational diabetes mellitus (GDM) is commonly diagnosed using an oral glucose tolerance test (OGTT) after 24 weeks of gestation [44]. This diagnosis is established using either a one-step or two-step approach with blood glucose thresholds recommended by the American Diabetes Association, the International Association of Diabetes in Pregnancy Study Groups (IADPSG), and the World Health Organization (WHO) [11]. However, despite its widespread use and effectiveness, the OGTT lacks clear consensus regarding optimal screening and diagnostic criteria for GDM [45,46,47]. Additionally, this method traditionally detects GDM at a relatively late stage of pregnancy when significant metabolic changes have already occurred, potentially limiting opportunities for and effectiveness of early interventions aimed at preventing the associated complications. Current evidence reveals that prolonged exposure of the fetus to maternal hyperglycemia prior to the diagnosis and treatment of GDM after the 24th week of gestation is linked to increased fetal growth and early onset of offspring adiposity [48,49]. This exposure also contributes to the longer-term cardio-metabolic risk observed in the offspring of women with GDM [19,50]. Consequently, there is growing interest in biomarkers that can identify the likelihood of developing GDM in pregnant women and predict the likely course or outcome of GDM after it has been diagnosed, ideally allowing for early intervention to prevent the onset of the disease or its complications. The primary objectives of this review are: first, to assess potential diagnostic biochemical markers for GDM; second, to evaluate prognostic biomarkers in pregnancies complicated by GDM; and, third, to identify potential predictors of long-term outcomes in women with GDM. We hypothesize that these biochemical markers may be useful for the early detection of GDM, the prediction of adverse pregnancy outcomes, and long-term outcomes.

## 2. Scope and Methodology

The scope of this review encompasses the diagnostic and prognostic value of biomarkers. For diagnostic studies, we examined markers from the first and second trimesters, extracting measures of accuracy. For prognostic studies, we considered all trimesters and focused on pregnancy complications and long-term outcomes in GDM.

A narrative review was conducted, following key steps in the review process as outlined below: ***(1) Review Question:*** The primary review question was to assess the usefulness of biomarkers in predicting the development of GDM and their role as prognostic factors for adverse pregnancy outcomes and long-term effects in women with GDM. ***(2) Search Strategy:*** A literature search was conducted in the PubMed database using the following terms: (Diagnos* OR Screen* OR Predict* OR Detect* OR “Pregnancy outcomes” OR Prognos*) AND (Biomarker OR Marker) AND (GDM OR “Gestational diabetes mellitus”). ***(3) Inclusion/Exclusion Criteria:*** The search was limited to original research articles of all designs, including in vivo studies in human subjects, published in English between 1 January 2014, and 30 June 2024, relevant to the research question, and identifying associations between biomarkers and GDM, pregnancy outcomes, or long-term prognosis. Studies lacking a clear definition of GDM, gestational age at testing, or standard laboratory assays of biomarkers were excluded. In vitro and animal studies were also excluded. ***(4) Critical Appraisal:*** Each study was assessed for quality by evaluating the robustness of its methods, the reliability of its findings, the clarity of its definitions, and its relevance to the review question. ***(5) Synthesis:*** Using a narrative approach, this review provides a coherent synthesis of the available literature by summarizing studies, identifying trends and emerging patterns, and assessing the consistency or discrepancies of findings across selected articles. It also examines the effect size of diagnostic values (sensitivity, specificity, and area under the curve), highlights gaps in knowledge and areas of controversy, and provides insights into how different studies relate to one another while suggesting future research directions. Contradictory findings are discussed, and all results were agreed upon by the authors. ***(6) Limitations:*** The limitations of this review are acknowledged and will be discussed.

In brief, a two-stage screening process was employed, first, to identify studies on biomarkers for GDM diagnosis and prognosis, and, second, to evaluate the evidence for each biomarker’s diagnostic performance and its potential prognostic correlation with GDM. This review highlights the existing knowledge gaps and proposes future research directions for discovering diagnostic and prognostic markers in GDM. Detailed information regarding this literature review is provided in the Appendix A.

## 3. Biochemical Markers for the Diagnosis of GDM

Several biomarkers may hold diagnostic potential, as they have been previously documented to be involved in the pathogenesis of GDM. Those diagnostic markers can be categorized into several groups: adipokines, inflammatory markers, insulin resistance markers, glycemic markers, placenta-derived markers, and other contextual markers, as follows.

### 3.1. Adipokines

Adipokines, secreted by adipose tissue, play roles in insulin sensitivity, glucose metabolism, inflammation, and lipid metabolism, making them potential biomarkers for predicting GDM. Adiponectin, an insulin-sensitizing adipokine, consistently shows decreased levels in first- and second-trimester GDM patients, with studies reporting an area under the curve (AUC) of 0.634–0.801 for its predictive ability [51,52,53,54,55]. In contrast, leptin shows elevated concentrations in GDM, with first-trimester levels achieving high sensitivity (95.7%) [51,56] and second-trimester measurements showing excellent overall performance (AUC 0.956) [57,58]. Chemerin, involved in adipocyte differentiation and glucose homeostasis, is increased in GDM. While its predictive value in the first trimester is modest (AUC 0.581), chemerin shows significantly improved performance in the second trimester with AUCs ranging from 0.820 to 0.970 [51,57,59]. Fatty acid-binding protein 4 (FABP4) consistently increases in GDM in both trimesters, with second-trimester measurements offering robust predictive power (AUC 0.814–0.940) [59,60,61]. Irisin, a myokine involved in energy metabolism, shows reduced concentrations in GDM, with first-trimester levels showing promising predictive potential (AUC 0.723–0.940) [62,63,64]. Retinol-binding protein 4 (RBP4), associated with insulin resistance, shows increased levels in GDM, with second-trimester measurements showing strong predictive performance (AUC 0.87) [65,66]. Betatrophin (ANGPTL8), involved in lipid metabolism, shows increased concentrations in GDM, with second-trimester levels offering good predictive value (AUC 0.812) [67,68].

In the first trimester, resistin appears to be a promising indicator with elevated levels (>5.3 ng/mL) showing a sensitivity of 95.7% and a specificity of 61.4% (AUC 0.836) [56]. In contrast, secreted frizzled-related protein 5 (SFRP5) shows a decrease in concentration, resulting in an AUC of 0.824 with a remarkable specificity (88.64%) [69]. Visfatin, which also shows an upward trend (>2.8 ng/mL), demonstrates strong sensitivity (87.1%) and moderate specificity (70%) [56]. C1q/TNF-related protein 9 (CTRP9) shows potential with an AUC of 0.776, although specific cutoff values and performance metrics require further clarification [70]. Fetuin-A and secreted frizzled-related protein 4 (SFRP4) show moderate predictive abilities (AUC 0.612 and 0.605, respectively) [51,71]. In the second trimester, asprosin appears to be an excellent marker with elevated levels (>31.709 ng/mL), providing exceptional sensitivity (93.3%) and specificity (90.9%), resulting in an AUC of 0.970 [72]. Fibroblast growth factor 21 (FGF-21) similarly shows excellent predictive power (AUC 0.95) [73], while lipocalin 2 (LCN2) [54] and omentin-1 show promising results with AUCs ranging from 0.836 to 0.887 [74,75]. Levels of C1q/TNF-related protein 3 (CTRP3) and myonectin (CTRP15) are lower in women with GDM, with myonectin (CTRP15) showing sensitivity (82.5%) and specificity (72.5%) [64,76]. Adropin, involved in glucose and lipid metabolism, also shows reduced levels with moderate predictive performance [77]. Pentraxin 3 (PTX3), an inflammatory marker, shows elevated concentrations in GDM, demonstrating high sensitivity but low specificity [78]. Neuregulin 4 (NRG4), involved in energy homeostasis, shows reduced levels with moderate predictive accuracy [79].

In summary, adipokine markers contributing to insulin resistance, such as leptin, chemerin, FABP4, and RBP4, are typically elevated in GDM, whereas those contributing to insulin sensitivity, like adiponectin, are reduced in GDM. The diagnostic value of most adipokines appears promising, although some inconsistencies have been observed.

### 3.2. Inflammatory Markers

GDM causes chronic low-grade inflammation, leading to elevated levels of inflammatory markers, including pro-inflammatory cytokines, such as IL-6 and TNF-alpha. Although these two markers have been extensively studied, research on their predictive performance remains limited. Changes in inflammatory markers may precede the onset of GDM and could serve as valuable indicators of disease risk and progression. Several key inflammatory markers have been extensively studied in the context of GDM. Interleukin-6 (IL-6) and tumor necrosis factor-alpha (TNF-alpha) show elevated levels in GDM cases, with IL-6 demonstrating moderate predictive power in the first trimester (AUC 0.673) [52,53,80]. While specific cutoff values and predictive abilities for TNF-alpha have not been reported, its consistent increase across trimesters suggests a potential role in the pathogenesis of GDM [53,81,82]. High-sensitivity C-reactive protein (hs-CRP) appears to be a particularly promising predictor, showing increased concentrations and strong diagnostic performance, especially in the second trimester, with AUC values ranging from 0.856 to 0.89 [74,75,83]. Fibrinogen, both a marker of inflammation and a central player in the coagulation cascade, is another promising marker in the second trimester. Elevated fibrinogen levels above 2.80 g/L were associated with increased GDM risk, demonstrating robust predictive capabilities with a sensitivity of 86.70%, specificity of 85.14%, and an AUC of 0.87 [74].

### 3.3. Insulin Resistance Markers

The homeostatic model assessment of insulin resistance (HOMA-IR) appears to be an important predictor, especially in the second trimester, with an AUC of 0.913, demonstrating high sensitivity (94.5%) and specificity (72.2%) [84,85,86]. This finding is consistent with the physiological insulin resistance that develops during pregnancy and is exacerbated in GDM. Sex hormone-binding globulin (SHBG) levels, which are reduced in GDM, show strong predictive performance in the both first and second trimesters, with AUCs of 0.874 and 0.897, respectively [87,88]. The triglyceride-glucose (TyG) index, a surrogate marker of insulin resistance, shows moderate predictive power in the first trimester (AUC 0.641–0.692) [84,89], reflecting the early metabolic disturbances of GDM. The new TyHGB marker, which is calculated as TG/HDL-C + 0.7 FBG (mmol/L) + 0.1 pre-pregnancy BMI (kg/m^2^), demonstrates similar moderate efficacy with 57% sensitivity and 70.3% specificity at a value of 6.16, resulting in an AUC of 0.682 in the first trimester [90]. The Quantitative Insulin Sensitivity Check Index (QUICKI), an insulin sensitivity marker, exhibits excellent performance in the second trimester (AUC 0.905) [86], underscoring the pivotal role of insulin sensitivity in GDM pathogenesis. C-peptide and insulin levels in the second trimester also show moderate predictive capacity (AUC 0.764 and 0.714, respectively) [91]. In summary, elevated HOMA-IR and decreased QUICKI, both indicative of reduced insulin sensitivity, appear to be diagnostic for GDM, although the number of studies is limited. Similarly, a decrease in SHBG is highly predictive, while C-peptide shows moderate diagnostic value.

### 3.4. Glycemic Markers

Glycemic markers, including glycated hemoglobin (HbA1c) and fasting plasma glucose (FPG), have been extensively studied as potential diagnostic markers for the development of gestational diabetes mellitus (GDM). Glycated hemoglobin (HbA1c) has demonstrated consistent performance in both trimesters. In the first trimester, HbA1c levels between 5.33 and 5.45% show good predictive capacity, with sensitivity ranging from 54.8% to 83.3% and specificity from 69.0% to 96.8% (AUC 0.809–0.840) [92,93,94]. This predictive ability was maintained in the second trimester, with slightly higher cutoff values of 5.45–5.7% yielding similar performance (AUC 0.826–0.848) [95,96]. Fasting plasma glucose (FPG) also demonstrated moderate predictive capacity, particularly in the first trimester, with levels between 81 and 88.5 mg/dL achieving sensitivities of 64.29–79.31% and specificities of 56.45–59.32% (AUC 0.63–0.738) [83,97,98]. Its performance in the second trimester remained consistent (AUC 0.712) [99]. In addition, a less-studied glycemic marker with comparable predictive performance is 1,5-anhydroglucitol (1,5-AG), a short-term glycemic marker reflecting postprandial glucose excursions. 1,5-Anhydroglucitol (1,5-AG), a marker of short-term glycemic control, showed promise in the second trimester with levels below 13.21 µg/mL demonstrating moderate predictive ability (AUC 0.693–0.722) [100]. Furthermore, one such novel glycemic marker is plasma glycated CD59 (pGCD59), which reflects background glucose levels. pGCD59 levels measured during gestation were significantly higher in women with GDM compared to those without GDM. Although pGCD59 levels exhibited limited predictive accuracy for GDM diagnosis later in pregnancy, with an AUC of 0.63–0.65, this marker demonstrated improved predictive capabilities in women with a higher BMI, achieving AUCs of 0.85 and 0.88 [101,102]. Additionally, second-trimester glycated albumin (GA) and fructosamine levels, which reflect average blood glucose levels over the past 2–3 weeks, had AUCs of 0.568 and 0.52 for predicting GDM, respectively [103,104]. In summary, among this group of biomarkers, only HbA1c appears promising and may serve as an early marker for the subsequent development of GDM.

### 3.5. Lipid Profile Markers

Lipid metabolism undergoes significant alterations during pregnancy, with a notable impact on the development of gestational diabetes mellitus (GDM). Among the various lipid markers, triglycerides (TG) have emerged as a promising predictor of GDM risk. Studies have consistently demonstrated elevated TG levels in women who subsequently develop GDM, with first-trimester TG concentrations ranging from 1.235 to 2.375 mmol/L, showing moderate to good predictive performance (sensitivity: 73.7–86.27%, specificity: 59.3–66.67%, AUC: 0.622–0.813) [105,106,107]. This predictive capacity persists into the second trimester, albeit with slightly different cutoff values (1.525–2.66 mmol/L) and comparable diagnostic accuracy (sensitivity: 72.09%, specificity: 71.6%, AUC: 0.587–0.769) [106,108]. The TG/HDL-C ratio, reflecting the balance between atherogenic and anti-atherogenic lipoproteins, has also demonstrated potential as a GDM predictor. First-trimester TG/HDL-C ratios of 0.831–2.2684 exhibit promising discriminatory power (sensitivity: 63.7–72.97%, specificity: 64.3–75.05%, AUC: 0.664–0.786) [84,109], while second-trimester values (1.12–4.254) show similar or slightly improved performance (sensitivity: 73.7–79.07%, specificity: 56.9–78%, AUC: 0.705) [108,110]. Remnant cholesterol (RC), a measure of cholesterol in triglyceride-rich lipoproteins, has shown excellent predictive ability in the first trimester at a cutoff of 24.30 mg/dL (sensitivity: 86.49%, specificity: 64.20%, AUC: 0.8038) [111]. The atherogenic index of plasma (AIP), calculated as log10(TG/HDL), offers another perspective on lipid imbalance, with a first-trimester cutoff of 0.3557 demonstrating good diagnostic accuracy (sensitivity: 72.22%, specificity: 75.41%, AUC: 0.7879) [112]. Interestingly, the LDL-C/HDL-C ratio, while altered in GDM, shows relatively modest predictive performance (sensitivity: 66.7%, specificity: 46.2%, AUC: 0.574) at a first-trimester cutoff of 0.928 [84]. In summary, all are likely to have diagnostic value, but their accuracy is inconsistent. In addition, the sample size in most studies was relatively small.

### 3.6. Placenta-Derived Markers

The placenta, a crucial organ in pregnancy, has garnered significant attention in the field of gestational diabetes mellitus (GDM) research. Placental growth factor (PlGF), with a median multiple of the median (MoM) of 0.89, demonstrated an upward trend in GDM cases, achieving a sensitivity of 51.2% and specificity of 87.2%, with an AUC of 0.68 [113]. This suggests that PlGF may play a role in the early placental dysfunction associated with GDM. In contrast, pregnancy-associated plasma protein-A (PAPP-A) and beta-human chorionic gonadotropin (β-hCG) exhibited downward trends. PAPP-A (MoM 0.885) showed a sensitivity of 66.67% and specificity of 65.50% (AUC 0.654), while β-hCG (MoM 0.990) presented a sensitivity of 74.40% and specificity of 46.80% (AUC 0.603) [114]. At 16–20 weeks, the second trimester introduces a new set of promising markers, including myostatin, soluble fms-like tyrosine kinase-1 (sFlt-1), follistatin (FST), and placental protein 13 (PP13). Notably, myostatin demonstrates exceptional sensitivity (92.5%) and a high AUC (0.84) in predicting GDM (104). sFlt-1, an anti-angiogenic factor, exhibits the highest sensitivity (94.9%) among the studied markers, although its specificity is relatively low [78]. FST and PP13 also show promise, with sensitivities exceeding 90% but varying specificities [78]. In summary, in terms of diagnostic value, markers from both the first and second trimesters are fairly diagnostic. However, an increase in myostatin shows relative promise, although only one study has been published.

### 3.7. Contextual Biochemical Markers

#### 3.7.1. Metabolic Markers

S(Pro)RR, a key component of the renin–angiotensin system, exhibits elevated levels in both the first and second trimesters, with high sensitivity and specificity (75–80% and 68–80%, respectively) [115,116]. Zonulin, a protein modulating intestinal permeability, also shows promise as a biomarker in the first and second trimesters, with notable predictive accuracy (AUC 0.755–0.796) [117,118]. The inverse relationship between 25-hydroxyvitamin D (25-(OH)D) levels and GDM risk is evident across trimesters, albeit with moderate predictive performance (AUC 0.66–0.721) [107,119,120]. Osteocalcin, involved in bone metabolism and glucose homeostasis, demonstrates a positive association with GDM in the first trimester; however, its predictive value appears limited (AUC 0.61) [121]. Second-trimester markers such as homocysteine (Hcy), vitronectin, and afamin offer additional insights. Elevated Hcy levels show high specificity (94.10%) but moderate sensitivity (62.70%) for GDM prediction [74]. Vitronectin, an adhesion protein, and afamin, a vitamin E-binding protein, both exhibit increased concentrations in GDM, with varying predictive accuracies (AUC 0.647 and 0.629, respectively) [79,122]. Overall, most of these markers have a moderate predictive capability in predicting GDM.

#### 3.7.2. Hematologic Markers

Hematological ratios derived from complete blood count parameters have shown potential in GDM prediction. The neutrophil-to-lymphocyte ratio (NLR) has moderate predictive ability in both trimesters, with improved performance in later pregnancy (AUC: 0.696–0.867) [123,124,125]. Similarly, novel inflammatory indices have also shown promise. The systemic immune-inflammation index (SII) and the systemic inflammatory response index (SIRI) have shown good predictive power across trimesters [75,126], with the SIRI, in particular, achieving an AUC of 0.833 in the second trimester [75]. The monocyte-to-lymphocyte ratio (MLR) and platelet-to-lymphocyte ratio (PLR) also show some predictive power, although with lower sensitivity and specificity compared to other markers [125]. Mean platelet volume (MPV) emerges as a particularly promising marker, demonstrating an upward trend in GDM cases across both the first and second trimesters. In the first trimester, MPV values ranging from 7.38 to 10.1 fL showed moderate predictive performance, with sensitivities of 50–69.5% and specificities of 65–66.3% (AUC 0.577–0.704) [127,128]. However, its predictive capacity improved significantly in the second trimester with values between 8.0 and 11.05 fL, achieving higher sensitivity (71.9–82%) and specificity (75–82.5%), resulting in an AUC of 0.805–0.906 [124,129]. This increase in predictive power suggests that platelet activation and dysfunction may become more prominent as GDM progresses. Plateletcrit (PCT) similarly exhibited an upward trend, with second-trimester measurements (0.19–0.20%) outperforming first-trimester values in terms of sensitivity (62.73–77.9%) and specificity (78.18–95%), yielding AUCs of 0.766–0.932 [127,130,131]. The neutrophil count also showed elevation in GDM, with first-trimester values (5.0–6.46 × 10^9^/L) demonstrating moderate predictive ability (AUC 0.63–0.655) [125,132]. Other hematological parameters, including platelet count, hemoglobin, lymphocyte count, and red blood cell count, displayed varying degrees of predictive capacity in the first trimester [132,133]. In the second trimester, white blood cell count (10.13 × 10^9^/L) also became a moderate predictor, with AUCs of 0.624 [125]. In brief, hematological markers exhibit moderate predictive capability, primarily belonging to hematological ratios such as the NLR, MLR, SIRI, and SII.

#### 3.7.3. Thyroid Function Markers

Thyroid-stimulating hormone (TSH) has shown promise as a biomarker, with elevated levels associated with an increased risk of GDM. In the first trimester, TSH levels above 5.33 mIU/L demonstrated moderate predictive capacity, with 62.9% sensitivity and 78.7% specificity (AUC 0.705) [134]. Interestingly, while second-trimester TSH cutoff values were lower (2.58 mIU/L), they maintained similar predictive performance (AUC 0.71), with improved specificity (92.4%) but reduced sensitivity (43.6%) [135]. This shift in cutoff values across trimesters highlights the dynamic nature of thyroid function during pregnancy and underscores the importance of trimester-specific reference ranges. Thyroid autoantibodies have also emerged as potential GDM predictors. Thyroid peroxidase antibodies (TPOAbs) have shown moderate predictive ability in both the first and second trimesters, with slightly better performance in the first trimester (AUC 0.642 vs. 0.665) [136]. Notably, thyroglobulin antibodies (TgAbs) have demonstrated superior predictive capacity, particularly in the second trimester, with an AUC of 0.833 [136]. This suggests that thyroid autoimmunity may play a significant role in GDM pathogenesis, possibly through shared autoimmune mechanisms or by influencing thyroid hormone levels. Free triiodothyronine (FT3) levels in the first trimester have shown good predictive ability (AUC 0.724) with a cutoff of 4.61 pmol/L, while second-trimester free thyroxine (FT4) levels below 14.00 pmol/L have demonstrated moderate predictive capacity (AUC 0.626) [136]. The opposing trends of FT3 and FT4 in relation to GDM risk suggest a complex interplay between thyroid hormones and glucose metabolism during pregnancy. These findings collectively indicate that thyroid dysfunction, characterized by elevated TSH levels, the presence of thyroid autoantibodies, and alterations in free thyroid hormone levels, may contribute to the development of GDM. In brief, among thyroid markers, elevated TgAb levels show promising predictive performance. However, the number of studies remains limited.

#### 3.7.4. Miscellaneous Markers

Serum ferritin (SF), an indicator of iron status, has shown notable associations with GDM risk. In the first trimester, elevated SF levels above 55.7 ng/mL demonstrated moderate predictive capacity, with a sensitivity of 62.6% and specificity of 53.7% [137]. The predictive performance of SF improved markedly in the second trimester, in which levels exceeding 37.55 ng/mL yielded an AUC of 0.904 [138]. These findings suggest that iron overload may contribute to the pathogenesis of GDM, potentially through increased oxidative stress and inflammation. In contrast, lactoferrin, an iron-binding glycoprotein with anti-inflammatory properties, has shown a significant decrease in GDM patients. First-trimester lactoferrin levels below 794.2 ng/mL have demonstrated exceptional predictive performance, with 100% sensitivity and 95.83% specificity, resulting in an AUC of 0.98 [139]. SERPINB1, a serine protease inhibitor involved in neutrophil function and inflammation, is indicative of increased GDM risk, with a sensitivity of 75.86%, specificity of 81.67%, and an AUC of 0.832 [79]. Lastly, cystatin-C (Cys-C), a marker of renal function and cardiovascular risk, has demonstrated moderate predictive value for GDM in the second trimester, yielding a sensitivity of 58.6%, specificity of 73.4%, and an AUC of 0.722 [140].

In summary, several biomarkers in all groups have the potential to predict the development of GDM as summarized in Figure 1. Among these, adipokine-related biomarkers have been studied most extensively, with several showing promise for early detection.

## 4. Biochemical Markers and Maternal Outcomes in Women with GDM

Elevated levels of fatty acid binding protein 4 (FABP4) in the second trimester were associated with an increased risk of pregnancy-induced hypertension (PIH), demonstrating an AUC of 0.647 [141]. Similarly, glycated hemoglobin (HbA1c) levels between 5.1 and 5.9% in the second trimester and above 5.9% in the third trimester were associated with a higher incidence of PIH [142,143]. Preeclampsia risk has been correlated with vitamin D deficiency, specifically with 25-hydroxyvitamin D (25OHD) levels below 10 ng/mL in the second trimester [144], as well as elevated serum ferritin concentrations exceeding 24.45 ng/mL [145]. First-trimester triglycerideglucose (TyG) index has shown potential in predicting gestational hypertension [89]. The risk of preterm delivery escalates with first-trimester HbA1c levels between 5.5 and 6.4% [146] and second-trimester levels above 5.5% [143,147]. Cesarean section rates increase with elevated second- and third-trimester triglyceride levels [148] and third-trimester glycated albumin levels above 15.69% [149,150]. Abnormal amniotic fluid volume has been associated with increased second-trimester HbA1c [96]. The risk of postpartum hemorrhage rises with elevated third-trimester glycated albumin [151]. Second-trimester HbA1c is correlated with increased shoulder dystocia risk [96], while elevated second-trimester growth differentiation factor 15 (GDF15) has been linked to a higher likelihood of microalbuminuria [152]. In conclusion, elevated HbA1c and reduced 25(OH)D levels are promising prognostic factors for predicting adverse outcomes in women with GDM.

## 5. Biochemical Markers and Fetal-Neonatal Outcomes in Women with GDM

Glycated hemoglobin (HbA1c) emerges as a consistent prognostic indicator throughout gestation, with elevated levels (≥5.7%) associated with an increased risk of large-for-gestational-age (LGA) infants and macrosomia [142,153,154,155]. The predictive ability of HbA1c for LGA, as measured by the AUC, ranges from 0.56 to 0.69 in the third trimester [142,154,155]. Triglyceride (TG) levels and the TG/HDL-C ratio also demonstrate significant predictive value, particularly in the second trimester, where a TG/HDL-C ratio cutoff of 1.85 combined with HbA1c and pre-pregnancy BMI yielded an AUC of 0.806 for LGA detection [110,148,156,157]. First-trimester FPG (82–92.3 mg/dL) and the triglyceride–glucose index (AUC: 0.643) are promising early LGA predictors [156,158,159]. Lower second-trimester 25-hydroxyvitamin D levels (<10 ng/mL) also correlate with increased LGA risk, suggesting the role of maternal vitamin D in fetal growth regulation [160]. Novel biomarkers such as plasma glycated CD59 (pGCD59) and glycated albumin (GA) show significant associations with LGA, with GA showing particularly high predictive power in the third trimester (AUC 0.80–0.92) [101,154,155,161,162]. For the prediction of macrosomia, first-trimester monocyte counts show an inverse relationship [163], while second- and third-trimester HbA1c levels ≥ 5.9% are positively correlated [96,142,143,164]. Glycated albumin, triglyceride, and fructosamine levels in later trimesters also increase the risk of macrosomia [148,149,165,166,167]. Notably, third-trimester FPG levels exceeding 98.6 mg/dL significantly elevates macrosomia risk [149]. In the second trimester, elevated HbA1c levels have been associated with an increased risk of neonatal hypoglycemia and the need for neonatal resuscitation [96]. Serum ferritin, with a cutoff value of 27.43 ng/mL, has demonstrated a strong predictive ability (AUC = 0.80) for neonatal hypoglycemia [145]. Moreover, elevated ferritin levels at both 16–18 and 28–32 weeks have been associated with increased risk of small-for-gestational age (SGA) infants [168]. The third trimester presents additional markers of interest, such as glycated albumin, which, at levels between 13.5% and 15.8%, shows a high predictive value (AUC: 0.79–0.90) for neonatal hypoglycemia [155,161,162]. In contrast, reduced levels of 25-OH-D (<20 ng/mL) in the third trimester have been linked to an elevated risk of both neonatal hypoglycemia and SGA infants [169]. Triglyceride levels in the second and third trimesters, as well as third-trimester zonulin concentrations, have been correlated with a higher likelihood of NICU admission [148,170]. Additionally, elevated second-trimester HbA1c and third-trimester glycated albumin (>13.7%, AUC: 0.53) have been identified as risk factors for neonatal hyperbilirubinemia [147,155]. Glycated albumin, measured in the third trimester, emerges as a significant predictor for multiple adverse outcomes, including neonatal hypocalcemia, polycythemia, respiratory disorders, and myocardial hypertrophy. Studies have reported different cutoff values ranging from 13.7% to 15.8%, with AUC values ranging from 0.61 to 0.92, indicating moderate to high predictive ability [155,161,162]. The GA/HbA1c ratio, with a cutoff value of 2.55 in the third trimester, also shows strong predictive ability for myocardial hypertrophy (AUC: 0.91) [155]. Zonulin, assessed in the third trimester, correlates with increased risk of low 1-min APGAR scores, low 5-min APGAR scores, and meconium-stained fluid, although specific cutoff values were not reported [117,170]. Vitamin D deficiency, characterized by 25OHD levels below 10 ng/mL in the second trimester, is associated with lower APGAR scores at both 1 and 5 min [144]. Other significant markers include an increased mean platelet volume (MPV) in the second trimester associated with a lower 1-min APGAR score [129] and elevated serum ferritin (SF) levels above 27.37 ng/mL in the second trimester, which correlate with a higher risk of respiratory disorders (AUC: 0.730) [145]. In summary, in women with GDM, glycated albumin is a promising prognostic factor for predicting large-for-date babies, myocardial hypertrophy, as well as neonatal hypoglycemia, hypocalcemia, and polycythemia.

## 6. Biochemical Markers and Long-Term Outcomes in Women with GDM

Elevated triglyceride levels in the second trimester, with a cutoff value of 2.89 mmol/L, have been associated with an increased risk of postpartum abnormal glucose metabolism (AGM) [171]. The presence of thyroid autoantibodies, specifically thyroid peroxidase antibodies (TPOAbs) and thyroglobulin antibodies (TgAbs), in the first trimester correlates with a higher incidence of postpartum glucose intolerance [136]. Hemoglobin A1c (HbA1c) levels exceeding 5.55% and 5.5% in the second trimester showed predictive value for both postpartum glucose intolerance and postpartum diabetes mellitus (PDM), with AUC values of 0.737 and 0.846, respectively [96,172]. Interestingly, prolactin levels below 115 ng/mL at the end of the second or beginning of the third trimester, as well as vitamin D concentrations below 20 ng/mL at 24–32 weeks, are associated with an increased risk of postpartum glucose dysregulation [173,174]. New evidence suggests that increased levels of pigment epithelium-derived factor (PEDF) in the second trimester may indicate a higher risk of postpartum prediabetes [175]. In addition, PEDF levels exceeding 4.23 μg/mL at 24–32 week showed strong predictive power for postpartum diabetes mellitus, with an AUC of 0.893 [176]. Betatrophin, measured in the second trimester, has also been implicated in the development of postpartum diabetes mellitus, although specific cutoff values have not been established [68]. In summary, with limited data, elevated levels of betatrophin, HbA1c, and PEDF are potential prognostic factors for predicting postpartum diabetes.

## 7. Challenges and Future Directions

Many biomarkers have recently emerged as potential indicators for GDM, offering a basis for early diagnosis and targeted treatment. Several biomarkers, including metabolites, microRNAs, and proteins, are implicated in insulin resistance, glucose metabolism, and inflammation. Thus, the complex network linking these biomarkers is highly informative for future clinical applications. However, based on this comprehensive review, no single biomarker has been sufficiently validated as an established screening tool for the early detection or prediction of GDM.

### 7.1. Limitations

The results from several studies remain inconsistent and potentially irreproducible. The limitations of existing research on diagnostic and prognostic biomarkers for GDM are mostly attributed to small sample sizes, demographic factors such as ethnicity, and the retrospective nature of case–control studies, which lack the rigor to provide accurate insights into the true incidence of GDM. One of the most significant limitations is that most studies have been conducted during the second trimester, primarily between 24–28 weeks, alongside routine screening for GDM. This timeframe is relatively late for studying the biochemical subclinical changes that occur long before clinical manifestations, whereas samples taken prior to the onset of GDM are more critical for early detection, prediction, and prevention. It is important to emphasize that GDM-associated biomarkers are often subtly released in early gestation, long before the manifestation of glucose impairment. Therefore, detecting these biomarkers may help identify women at high risk for GDM or in the early stages of the condition. Notably, the results synthesized in this review may not be perfectly reliable due to the nature of narrative reviews, which are more prone to bias compared to systematic reviews, as they rely on the subjective selection and interpretation of studies. Finally, another limitation of this review is that it was based solely on the PubMed database, which may have led to the omission of some articles from other databases. Nevertheless, it is unlikely that significant studies on this topic are not covered by PubMed.

### 7.2. Gap of Knowledge

Translating insights from basic scientific knowledge of biochemical markers associated with the development of GDM to their clinical application requires further elucidation in several aspects. For example: (1) Although several biomarkers appear promising, no single biomarker has been sufficiently validated, either internally or externally, as an effective clinical predictor of GDM. (2) Studies evaluating the usefulness of such biomarkers during the first trimester, which is crucial for early detection, are still very limited. (3) There is a lack of research on longitudinal changes in biomarkers throughout pregnancy. (4) The combination of various promising biomarkers to improve diagnostic performance as part of a predictive model is still lacking. (5) The cost–benefit analysis of the clinical application of these biomarkers has not yet been evaluated. (6) Additionally, several emerging biomarkers linked to the pathogenesis of GDM, particularly in the field of metabolomics, have yet to be assessed for their diagnostic performance.

### 7.3. Future Directions

Accordingly, prospective studies with larger sample sizes and more diverse populations are needed to explore the associations between promising biomarkers and GDM, particularly focusing on the first trimester or early second trimester, before the development of glucose impairment. Importantly, the development of prediction models derived from multiple biomarkers is essential for improving diagnostic and prognostic performance. Furthermore, a systematic review and meta-analysis of the diagnostic performance of various promising markers, which are scattered across multiple studies, should be conducted. Additionally, the feasibility of implementing these markers in clinical practice needs to be evaluated. Moreover, long-term follow-up in prospective studies is essential to assess the utility of these biomarkers as screening tools and to evaluate their acceptability among women. Finally, given that GDM increases the risk of T2D later in life, research on postpartum biomarker tests for predicting T2D after GDM, with extended follow-up periods, is encouraged, as current studies are often limited to specific ethnic groups and short-term follow-up.

To date, although a significant number of studies on biomarkers associated with GDM have been conducted, and substantial insights have been gained over the past decades, the pathophysiology and pathogenesis of GDM remain incompletely understood. The identification of various biomarkers and the integration of omics-based evidence are crucial and hold promise for future research directions, despite the challenges involved. It is hoped that this comprehensive review will pave the way for future research on biomarkers, which can guide the early detection or prevention of GDM.

## 8. Supplementary Data 

The sources of data are summarized in Appendix A. 

## 9. Conclusions

This review provides a comprehensive summary of contemporary evidence on biomarkers, focusing on their potential to predict the development of GDM, facilitate early detection, and serve as predictors of maternal, fetal, and neonatal outcomes in women with GDM. Adipokines, inflammatory markers, insulin resistance markers, glycemic markers, lipid profile markers, and placenta-derived markers have shown promise in identifying women at risk of developing GDM and predicting adverse pregnancy outcomes. This review highlights promising biomarkers for future validation, as well as their potential combinations to improve predictive performance. To date, there has been no single biomarker demonstrating sufficient accuracy to replace the current diagnostic criteria for GDM. The complexity of multiple pathways in GDM pathogenesis highlights the need for a multi-marker approach to improve risk stratification and guide personalized management strategies. In conclusion, while the field of GDM biomarker research has made significant progress, further studies are needed to refine and validate these markers for clinical use and to develop a comprehensive, evidence-based approach to GDM prediction and management that can improve maternal and child health outcomes.

## Figures and Tables

**Figure 1 ijms-25-11666-f001:**
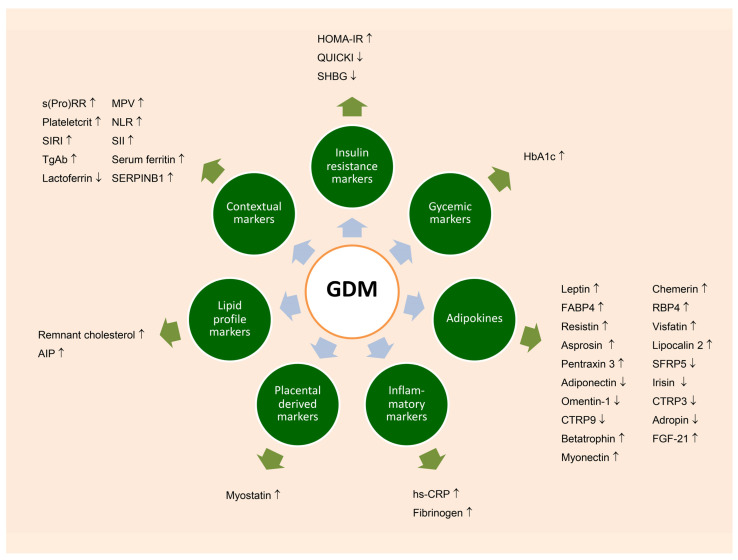
Biochemical markers in various groups associated with GDM, which appear to be promising predictors for subsequent development of GDM.

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
