# Peer review of "Predictive Utility of Biochemical Markers for the Diagnosis and Prognosis of Gestational Diabetes Mellitus"

_ijms, 2024, doi:10.3390/ijms252111666_

Round 1

Reviewer 1 Report

Comments and Suggestions for Authors

The article titled " Predictive utility of biochemical markers for the diagnosis and prognosis of gestational diabetes mellitus" by Sathaphone Inthavong et al. presents valuable insights; however, there are several areas that require attention:

1. It is important that the authors provide their own justification since the relevance of the study has already been explored in various literature and clinical studies in PubMed: Medicina (Kaunas). 2024 Jul 31;60(8):1250. doi: 10.3390/medicina60081250; Cardiovasc Diabetol. 2019 Oct 30;18(1):140. doi: 10.1186/s12933-019-0935-9; Int J Mol Sci. 2021 May 24;22(11):5512. doi: 10.3390/ijms22115512; EBioMedicine. 2024 Mar;101:105008. doi: 10.1016/j.ebiom.2024.105008; Curr Diab Rep. 2017 Feb;17(2):12. doi: 10.1007/s11892-017-0834-y; and hence I do not find any innovative information in the study

2. The authors can create a comprehensive table listing the biochemical markers used for diagnosing GDM. This table should include evidence from in vitro or in vivo experimental models, clinical evidence (if available), methods for validation, up or downregulated biochemical parameters during GDM, and corresponding references

3. While the authors mentioned the multi-marker approach and its importance, they did not provide specific recommendations on how combining biomarkers can enhance predictive accuracy and clinical effectiveness.

4. The authors should engage in a thorough discussion analyzing the gaps and limitations present in the study.

5. It is necessary for the authors to incorporate future research directions, recommendations, and potential study limitations.

6. The authors must include a flowchart or schematic diagram that illustrates the process of identifying, evaluating, and clinically applying biomarkers in the context of GDM.

Comments on the Quality of English Language

Moderate English corrections (grammatical, stylistic, and typographical) are required

Author Response

Predictive utility of biochemical markers for the diagnosis and prognosis of gestational diabetes mellitus

Reviewer #1

Comments and Suggestions for Authors

The article titled " Predictive utility of biochemical markers for the diagnosis and prognosis of gestational diabetes mellitus" by Sathaphone Inthavong et al. presents valuable insights; however, there are several areas that require attention:

  1. It is important that the authors provide their own justification since the relevance of the study has already been explored in various literature and clinical studies in PubMed: Medicina (Kaunas). 2024 Jul 31;60(8):1250. doi: 10.3390/medicina60081250; Cardiovasc Diabetol. 2019 Oct 30;18(1):140. doi: 10.1186/s12933-019-0935-9; Int J Mol Sci. 2021 May 24;22(11):5512. doi: 10.3390/ijms22115512; EBioMedicine. 2024 Mar;101:105008. doi: 10.1016/j.ebiom.2024.105008; Curr Diab Rep. 2017 Feb;17(2):12. doi: 10.1007/s11892-017-0834-y; and hence I do not find any innovative information in the study.

Response 1: Certainly, there are no new findings derived from this study, as it is not an original research article. However, this review provides an updated perspective on the topic through a systematic categorization of various biomarkers from numerous studies scattered throughout the literature. This approach makes it easier to understand the overall landscape and highlights which biomarkers show promise and are suitable for further investigation, paving the way for future research directions in this area.

Though many reviews have been published, our review focuses differently on biomarkers as a diagnostic or prognostic tools of GDM, rather than pathogenesis, focusing on translational researches rather than purely basic sciences. Actually, to the best of our knowledge, this is the first review providing a comprehensive summary of contemporary evidence on biomarkers, focusing on their potential to predict the development of GDM, and prognostic predictors of outcomes in women with GDM.

In this review, opinions on diagnostic or prognostic performance, though based on our own justification, they are derived from evidence from various studies, agreement across various studies and consensus by the authors’ team. We add in the “Methods” that any opinions are drawn from those studies. Some inconclusive, promising or less valuable markers are drawn from the evidence and our consensus in some contradictory results among various studies. This goes along with the typical process of narrative review.

  1. The authors can create a comprehensive table listing the biochemical markers used for diagnosing GDM. This table should include evidence from in vitro or in vivo experimental models, clinical evidence (if available), methods for validation, up or downregulated biochemical parameters during GDM, and corresponding references

Response 2: In revised MS, we clearly state in “Method” Inclusion/exclusion that this review include only in vivo studies relevant to the review questions and presented as categorized by the groups of markers according to pathogenesis.

  1. While the authors mentioned the multi-marker approach and its importance, they did not provide specific recommendations on how combining biomarkers can enhance predictive accuracy and clinical effectiveness.

Response 3: Thank you very much for the comments. In revised MS, we add a comment on combination of biomarkers in “Future directions”, as highlighted.

  1. The authors should engage in a thorough discussion analyzing the gaps and limitations present in the study.

Response 4: In revised MS, we have engage in a thorough discussion and analyzing the gap as presented in subheading 7.

  1. It is necessary for the authors to incorporate future research directions, recommendations, and potential study limitations.

Response 5: In revised MS, we incorporate future research directions and recommendations as presented in subheading 7 and potential study limitations in subheading 7.

  1. The authors must include a flowchart or schematic diagram that illustrates the process of identifying, evaluating, and clinically applying biomarkers in the context of GDM.

Response 6: In revised MS, a schematic diagram is added, as presented in Figure 1.

Comments on the Quality of English Language

Moderate English corrections (grammatical, stylistic, and typographical) are required

Response: The revised MS is now re-checked and corrected for English by a professional English speaker.

Reviewer 2 Report

Comments and Suggestions for Authors

The manuscript submitted for evaluation is a valuable review of the literature on biochemical factors as potential predictors in the diagnosis and prognosis of gestational diabetes mellitus. Below are my comments:

1.      The introduction to the manuscript describes the issues discussed well and provides a good overview. However, it does not clearly state the purpose of the study.

2.      literature reviews are usually performed based on a review of several article databases, here the authors chose only the PubMed database. A well-prepared review should be conducted in at least 3 databases, which will ensure that all studies are found.

3.      the inclusion and exclusion criteria were not clearly and explicitly specified in the methodology chapter

4.      it is not clear to me whether the authors conducted an assessment of the value and reliability of the selected works? The scientific level of the works included in the database is different, some of them (due to their nature) may contain methodological limitations or systematic or directional errors.

5.      it is not clear to me whether the authors conducted an assessment of the value and reliability of the selected articles? The scientific level of the papers included in the database is different, some of them (due to their nature) may contain methodological limitations or systematic or directional errors. Even in this database we can find works examining, for example, only the group with GDM and referring to the reference results in general or patients without GDM from other publications.

6.      Wouldn't it be more appropriate to conduct the work as a systematic review? In the individual subchapters, the authors do not go into details related to the mechanism of action of individual factors, but generally categorize them. In such a case, it would be necessary to search more databases and also add a PRISMA diagram.

7.      This will also allow for a meta-analysis, which is the only one that will allow for determining the importance of a given factor as a predictor. The value of the presented work will also be greater.

8.      I miss a graphic summary or a summary table in the body of the article, the table is in the supplement, but in the article it should be found as a summary collecting the information presented by the authors. It will significantly increase the value of the work.

Author Response

Predictive utility of biochemical markers for the diagnosis and prognosis of gestational diabetes mellitus

Reviewer #2

Comments and Suggestions for Authors

The manuscript submitted for evaluation is a valuable review of the literature on biochemical factors as potential predictors in the diagnosis and prognosis of gestational diabetes mellitus. Below are my comments:

  1. The introduction to the manuscript describes the issues discussed well and provides a good overview. However, it does not clearly state the purpose of the study.

Response 1: In revised MS, the purpose is now clearly stated at the end of “Introduction”, as suggested, as highlighted.

  1. literature reviews are usually performed based on a review of several article databases, here the authors chose only the PubMed database. A well-prepared review should be conducted in at least 3 databases, which will ensure that all studies are found.

Response 2: Given the large number of studies on this topic and the comprehensive coverage of high-impact medical journals by PubMed, we believe it adequately includes articles relevant to our research. It is unlikely that significant studies on this topic are not covered by PubMed. We respectfully request that the use of only the PubMed database be accepted, as commonly seen in several high standard journals. However, we acknowledge this limitation in our review, noting that 'One limitation of this review is that studies published in journals not indexed by PubMed are not included.' Nevertheless, if the editor or reviewer strongly recommends the inclusion of at least two additional databases, we are willing to comply.

  1. the inclusion and exclusion criteria were not clearly and explicitly specified in the methodology chapter

Response 3: In revised MS, the exclusion and inclusion criteria are now clearly stated in “Methodology”, as highlighted. (Inclusion: original articles, in vivo study, conducted on pregnant women; Exclusion: animal study)

  1. it is not clear to me whether the authors conducted, an assessment of the value and reliability of the selected works? The scientific level of the works included in the database is different, some of them (due to their nature) may contain methodological limitations or systematic or directional errors.

Response 4: In revised MS, the assessment of the value and reliability of the selected works are now clearly stated in “Methodology”, as highlighted.

Assessments are based on consistency across various studies, the unclear-cut issue (inconclusive) based on our consensus but all are based on evidence. The promising biomarkers are based on their consistency or agreement among studies, effect size based on diagnostic index or area under curve and all opinions are consensus among the author team.

The key steps in this review are as follows:

  1. Review Question: The primary review question is to assess the usefulness of biomarkers in predicting the development of GDM and as a prognostic factors for adverse pregnancy outcomes and long-term effects among women with GDM.
  2. Search Strategy: A literature search was conducted in the PubMed database using the following terms: (Diagnos* OR Screen* OR Predict* OR Detect* OR "Pregnancy outcomes" OR Prognos*) AND (Biomarker OR Marker) AND (GDM OR "Gestational diabetes mellitus").
  3. Inclusion/exclusion criteria: The search was limited to original research articles of all designs, in vivo studies in human subjects, published in English between January 1, 2014, and June 30, 2024, relevant to the research question, and indicating the association between biomarkers and GDM, pregnancy outcomes, or long-term prognosis. Studies lacking a clear definition of GDM, gestational age at testing, and standard laboratory assays of biomarkers were excluded. The in vitro studies and animal studies were also excluded.
  4. Critical appraisal: Each study was assessed for quality by evaluating the robustness of its methods, reliability of findings, clarity in defining terms, and relevance to the review question.
  5. Synthesis: This review provides a coherent synthesis of the available literature by summarizing studies, identifying trends, and assessing the consistency of findings across selected articles. It also examines the effect size of diagnostic values (sensitivity, specificity, area under the curve), highlights gaps in knowledge or areas of controversy, and provides insights into how different studies relate to one another while suggesting future research directions. Contradictory findings are discussed, and all results are agreed upon by the authors.
  6. Limitations: The limitations of this review are acknowledged and will be discussed.

  1. it is not clear to me whether the authors conducted an assessment of the value and reliability of the selected articles? The scientific level of the papers included in the database is different, some of them (due to their nature) may contain methodological limitations or systematic or directional errors. Even in this database we can find works examining, for example, only the group with GDM and referring to the reference results in general or patients without GDM from other publications.

Response 5: In revised MS, the assessment of the value and reliability of the selected works  are now clearly stated in “Methodology”, as highlighted.

  1. Wouldn't it be more appropriate to conduct the work as a systematic review? In the individual subchapters, the authors do not go into details related to the mechanism of action of individual factors, but generally categorize them. In such a case, it would be necessary to search more databases and also add a PRISMA diagram.

Response 6: We conducted a narrative review to assess all updated markers rather than a systematic review or meta-analysis, which is to focus on some specific biomarkers, as the aim of our review was to update and identify new, promising biomarkers. We respectfully request that a systematic review, which would require a PRISMA diagram, is not necessary for this study. Although this review primarily focuses on the diagnostic and prognostic value of biomarkers with the hope of our result may help to guide for future systematic review, as we mentioned in revised MS in part of future direction.

  1. This will also allow for a meta-analysis, which is the only one that will allow for determining the importance of a given factor as a predictor. The value of the presented work will also be greater.

Response 7: In the revised manuscript, we provide an overview of all promising biomarkers, in line with the nature of a narrative review, which focuses on diagnostic and prognostic insights to update current evidence without addressing specific questions as a meta-analysis does. Nonetheless, our comprehensive review can serve as a guide for scientists to conduct future meta-analyses on individual or groups of biomarkers. In the revised manuscript, we suggest future directions for clinical applicability through meta-analyses. However, for this study, we kindly request to present it as a narrative review, offering an overall perspective on the diagnostic and prognostic values of the biomarkers.

  1. I miss a graphic summary or a summary table in the body of the article, the table is in the supplement, but in the article it should be found as a summary collecting the information presented by the authors. It will significantly increase the value of the work.

Response 8: In revised MS, we provide graphic summary of diagnostic value based on the promising evidence (Figure 1).

Round 2

Reviewer 1 Report

Comments and Suggestions for Authors

Accept in present form

Author Response

Thank you very much

Reviewer 2 Report

Comments and Suggestions for Authors

The authors responded to all my comments and made satisfactory corrections to most of them. The only remaining issue is the number of databases in which the search is conducted. In my opinion, articles based on 1 database will always leave a presumption of incompleteness and will be assessed worse by potential readers. Moreover, if I had to choose as a citation for my article the one based on 1 database and the one based on several, I will choose the second one because it is more methodologically correct for me and there is no doubt that important aspects have been omitted. Even though they would describe the discussed issue in a similar way and at an equal level. I leave the decision of whether or not to add other databases to the search to the Editor - in accordance with the requirements and policy of the journal.

Author Response

Predictive utility of biochemical markers for the diagnosis and prognosis of gestational diabetes mellitus

Reviewer #2

Comments and Suggestions for Authors

The authors responded to all my comments and made satisfactory corrections to most of them. The only remaining issue is the number of databases in which the search is conducted. In my opinion, articles based on 1 database will always leave a presumption of incompleteness and will be assessed worse by potential readers. Moreover, if I had to choose as a citation for my article the one based on 1 database and the one based on several, I will choose the second one because it is more methodologically correct for me and there is no doubt that important aspects have been omitted. Even though they would describe the discussed issue in a similar way and at an equal level. I leave the decision of whether or not to add other databases to the search to the Editor - in accordance with the requirements and policy of the journal.

Response:

We have reviewed the author's guidelines for the International Journal of Molecular Sciences (IJMS) for review articles, which do not require the use of a specific number of databases. We believe that our narrative review meets the journal's requirements. Given the abundance of studies on this topic, PubMed comprehensively covers all significant medical journals and adequately includes articles relevant to our research. It is unlikely that significant studies on this topic would not be indexed in PubMed. Therefore, we respectfully request acceptance of the use of only the PubMed database, which is a common practice in several high-standard journals, including IJMS (for examples: doi.org/10.3390/ijms25115903; doi.org/10.3390/ijms232112727; doi.org/10.3390/ijms23169431). Moreover, many reviews do not describe their search methods in detail.

However, in the second revised manuscript, we have addressed this limitation in the "Limitations" subsection of item 7 (Challenges and Future Directions), as highlighted.